# Less Computer Access: Is It a Risk or a Protective Factor for Cyberbullying and Face-to-Face Bullying Victimization among Adolescents in the United States?

**DOI:** 10.3390/bs13100834

**Published:** 2023-10-11

**Authors:** Jun Sung Hong, Miao Wang, Rekha Negi, Dexter R. Voisin, Lois M. Takahashi, Andre Iadipaolo

**Affiliations:** 1School of Social Work, Wayne State University, Detroit, MI 48202, USA; fl4684@wayne.edu (J.S.H.); andre.iadipaolo@wayne.edu (A.I.); 2Department of Social Welfare, Ewha Womans University, Seoul 03760, Republic of Korea; 3Zhou Enlai School of Government, Nankai University, Tianjin 300350, China; wangmiao8609@163.com; 4Jesus and Mary College, University of Delhi, New Delhi 110021, India; rekhajsnegi@yahoo.com; 5Bullying Research Network, College of Education and Human Sciences, University of Nebraska-Lincoln, Lincoln, NE 68583, USA; 6Jack, Joseph and Morton Mandel School of Applied Social Sciences, Case Western Reserve University, Cleveland, OH 44106, USA; 7Price School of Public Policy, University of Southern California, Los Angeles, CA 90089, USA; lmtakaha@usc.edu

**Keywords:** adolescents, bullying, computer, cyberbullying, socioeconomic status, victimization

## Abstract

The present study investigates whether less computer access is associated with an increase or decrease in cyberbullying and face-to-face bullying victimization. Data were derived from the 2009–2010 Health Behavior in School-Aged Children U.S. Study, consisting of 12,642 adolescents aged 11, 13, and 15 years (Mage = 12.95). We found that less computer usage was negatively associated with cyberbullying victimization and face-to-face bullying victimization. The findings from the study have implications for research and practice.

## 1. Introduction

Despite the widespread implementation of anti-bullying programs and policies in U.S. school districts, bullying continues to be a severe concern for students, parents, teachers, and school officials. Being able to identify bullying and its many forms is essential for educators and practitioners to address and prevent student bullying. According to the Centers for Disease Control and Prevention, bullying is “an unwanted, aggressive behavior by… youth or [a] group of youths that involves an observed or perceived power imbalance” [1] (p. 7). Bullying is typically “repeated multiple times or is highly likely to be repeated” [1] (p. 7) and can be “direct” and “indirect”. Direct bullying consists of behaviors that occur “in the presence of the targeted youth”, like physical violence or verbal harassment. Indirect bullying consists of actions outside the targeted students’ presence, such as online harassment [1].

Over the years, empirical research has investigated individual-level antecedents of adolescents’ face-to-face bullying and cyberbullying victimization, such as sex, age, and race/ethnicity. There are inconsistent findings on sex differences. However, numerous studies show that males tend to be more prone to face-to-face bullying [2,3], whereas females tend to have a higher risk of indirect forms of bullying, including cyberbullying victimization [4]. Regarding age, research points out that bullying peaks in early adolescence [5] and gradually decreases as adolescents age [6,7]. Research findings on race/ethnicity and bullying have also been inconsistent. Some studies report that Whites report bullying victimization more frequently than racial and ethnic minorities [8], whereas other scholarly findings suggest that minorities have a higher risk [9,10].

### 1.1. Income Inequity and Bullying

The income inequality hypothesis is that there is a positive linkage between deprivation and children’s bullying victimization. This hypothesis proposes that poverty and deprivation further inequality, eroding social capital and contributing to social problems, including interpersonal violence [11,12]. Also, deprivation represents the inability to access material goods and services, which are essential components of functioning as a member of society (e.g., having internet access) [11,13], potentially increasing the risk of bullying victimization.

Most, but not all, empirical research supports the income inequality hypothesis, showing a positive association between low SES and resource deprivation and face-to-face bullying victimization. Wilson et al.’s [14] study found that economic deprivation was not significantly related to children’s bullying victimization. A meta-analytic review of the research on SES and bullying further found that although SES was weakly related to students’ bullying roles, victims and perpetrators of bullying were less likely to come from high SES backgrounds [15].

However, the bulk of research studies support the income inequality hypothesis. According to Glew et al. [16], bullying victimization, primarily face-to-face bullying victimization, is positively associated with poverty status, as measured by eligibility for free school lunches. A more recent study by Woolweaver et al. [17], which analyzed data from 20,302 high school students in Wisconsin, found that low financial status was a salient predictor of bullying victimization. 

Other studies explored whether economic disadvantage and hardship might be related to children’s involvement in face-to-face bullying and cyberbullying [18,19,20]. In a survey of 162,305 students from 5998 schools in 35 countries in Europe and North America, Due and colleagues [19] found that students of socioeconomic disadvantage had a higher prevalence of bullying victimization than those of higher SES. More generally, resource deprivation is associated with bullying victimization. Several studies have documented that children who suffer from material deprivation are likely to be psychologically distressed, less popular among peers, display behavioral problems, and have suicidal thoughts [21,22,23]. In turn, deprivations, particularly material deprivation [24,25], food insecurity [26,27], and relative deprivation [11] are significantly and positively associated with bullying victimization. Impoverished children are likely to be deprived of resources, which increases their risk of bullying victimization [28].

### 1.2. Cyberbullying

An increasingly important type of bullying is cyberbullying. Adolescents are avid internet users [29], and cyberbullying has become a significant public health concern in the last few decades. Cyberbullying victimization mainly occurs outside the school and in cyberspace [7], which requires access to an electronic device (e.g., a computer or a phone) [30]. Access to the internet requires access to computers, tablets, or smartphone technology. Existing research has shown that resource deprivation is associated with an increased risk of bullying victimization. However, with cyberbullying, a lack of access to the material resources needed for internet access may act as a protective factor that minimizes the risk of cyberbullying. 

To date, the bulk of empirical studies have focused on whether poverty and deprivation are related to children’s experiences of face-to-face bullying. Additionally, research has not considered whether aspects of deprivation, such as a lack of computer access, for example, might be protective against bullying and cyberbullying, as deprivation can shield youth from having contact with bullies. 

### 1.3. The Present Study

To address the issue of resource deprivation as a protective factor in cyberbullying victimization, the present study explores whether having limited access to electronic devices, namely a computer, is associated with an increase or decrease in bullying victimization. The following hypotheses are tested:
**Hypothesis** **(H1):***Less computer usage is associated with a decreased risk of cyberbullying victimization.*
**Hypothesis** **(H2):***Less computer usage is related to an increased risk of face-to-face bullying victimization.*

## 2. Materials and Methods

### 2.1. Sample and Data

Data for the present study were derived from the 2009–2010 Health Behavior in School-Aged Children (HBSC) U.S. Study, which contains the most recent data that were collected in the United States. The HBSC is a standardized, international research project conducted by the World Health Organization. The HBSC consists of repeated cross-sectional surveys conducted in the 43 participating countries through school-based surveys utilizing random sampling to choose a proportion of adolescents who are 11, 13, and 15 years of age [31]. The school-based survey comprises a self-reported questionnaire, which was completed by students from public schools. Students were administered the questionnaire in their classrooms. The questionnaire consists of a range of health indicators and health-related behaviors, along with questions on the students’ life circumstances [32]. Questions in the survey covered sociodemographic characteristics (e.g., age, gender), social background, social context, health outcomes, health behaviors, and risk behaviors [32].

### 2.2. Measures

Cyberbullying victimization was measured with four items. Respondents were asked the following question: “How often have you been bullied at school in the past couple of months in the ways listed below? (a) I was bullied using a computer or e-mail messages or pictures; (b) I was bullied using a cell phone; (c) I was bullied outside of school using a computer or e-mail messages or pictures; (d) I was bullied outside of school using a cell phone”. Response options were as follows: I haven’t been bullied at school in the past couple of months (0), It has only happened once or twice (1), 2 or 3 times a month (2), about once a week (3), and several times a week (4) (α = 0.92). The sum score of the four items was calculated to measure cyberbullying victimization. 

Face-to-face bullying victimization was measured with four items that were combined and summed. The items were “How often have you been bullied at school in the past couple of months in the ways listed below? (a) I was called mean names, was made fun of, or teased in a hurtful way; (b) other students left me out of things on purpose, excluded me from their group of friends, or completely ignored me; (c) I was hit, kicked, pushed, shoved around, or locked indoors; (d) others told lies or spread false rumors about me and tried to make others dislike me”. Response options for the item were as follows: I haven’t been bullied at school in the past couple of months (0), It has only happened once or twice (1), 2 or 3 times a month (2), about once a week (3), and several times a week (4) (α = 0.81). 

A lesser extent of computer usage was measured with four items, which included the following questions: (a) “About how many hours a day do you usually play games on a computer or game console (Playstation, Xbox, GameCube, etc.) in your free time? (Please mark one circle for weekdays)”; (b) “About how many hours a day do you usually play games on a computer or game console (Playstation, Xbox, GameCube, etc.) in your free time? (Please mark one circle for the weekend)”; (c) “About how many hours a day do you usually use a computer for chatting online, internet, emailing, homework, etc., in your free time? (Please mark one circle for weekdays)”; (d) “About how many hours a day do you usually use a computer for chatting online, using the internet, emailing, homework, etc., in your free time? (Please mark one circle for the weekend)”. Response options were as follows: none at all (0), about half an hour a day (1), about 1 h a day (2), about 2 h a day (3), about 3 h a day (4), about 4 h a day (5), about 5 h a day (6), about 6 h a day (7), and about 7 or more hours a day (8). All four items were reverse-coded and combined, and the sum score of recoded values reflected the lesser extent of computer usage (α = 0.74). 

Covariates were sex, age, race, ethnicity, family’s financial well-being, and parents’ occupations. Sex was measured with the question, “Are you a boy or a girl?” The response options were boy (1) and girl (2). Age was measured with the question, “How old are you?”. The response options ranged from 10 or younger (1) to 17 or older (7). Race was measured with the question, “What do you consider your race to be?” The response options were Black/African American (1), White (2), Asian (3), American Indian/Alaska Native (4), Native Hawaiian/Other Pacific Islander (5), and Other (6). Ethnicity was measured with the question, “What do you consider your ethnicity to be?” The response options were not Hispanic/Latino (0) and Hispanic/Latino (1). The family’s financial well-being was measured with the question, “How well off do you think your family is?” The response options were very well off (1) to not at all well off (5). Parents’ occupations were measured with two separate questions, “Does your father have a job?” and “Does your mother have a job?” The response options were no (0) and yes (1), both parents employed (2), only one parent employed (3), and both parents unemployed (4). 

### 2.3. Analytic Techniques

A descriptive statistical analysis was conducted to explore the distributions of the study variables and describe the study sample. Frequency and percentage values were included for discrete variables, and mean and standard deviation values were reported for the continuous variables. Linear regression analyses were then conducted to investigate the impacts of a lesser extent of computer usage on cyberbullying and face-to-face bullying victimization while controlling for sex, age, race, ethnicity, family’s financial well-being, and parents’ occupations. All data were analyzed using SPSS 20.0 [33]. 

## 3. Results

### 3.1. Descriptive Statistics

A total of 12,642 survey questionnaires were received and analyzed. The mean prevalence of cyberbullying victimization was 4.58 (*SD* = 2.18), and for face-to-face bullying victimization, it was 5.84 (*SD* = 3.15), with a range of 4.0 to 20.0. The mean for individuals with a lesser extent of computer usage was 27.38 (*SD* = 6.34), with a range of 4.0 to 36.0. 

Among all participants, 48.6% were girls, 51.4% were boys, and the average age was 12.95 years. Regarding race and ethnicity, 3407 participants were Hispanic/Latino (28.7%), and 2562 were Black/African American (20.3%). Most participants (52.1%) were reported as being White, while another 5.4%, 5.1%, and 1.8% were Asian, American Indian/Alaska Native, and Native Hawaiian/other Pacific Islanders, respectively. The mean of the family’s financial well-being was 2.54, indicating that the average family was quite well off. For parental occupations, 9547 fathers had a job (90.5%), while only 68.6% of mothers had a job (*n* = 8672). All information is presented in Table 1.

### 3.2. Regression Analysis

We computed a linear regression analysis to examine the effects of less computer usage on cyberbullying and face-to-face bullying victimization with two models. As shown in Table 2, a lesser extent of computer usage was negatively associated with cyberbullying victimization (*B* = −0.086, *p* < 0.001), controlling for the demographic variables. In addition, being Hispanic/Latino (*B* = −0.050, *p* < 0.001) and being Asian (*B* = −0.028, *p* < 0.05) were negatively associated with cyberbullying victimization. Being a girl (*B* = 0.022, *p* < 0.05), being Black/African American (*B* = 0.039, *p* < 0.01), having a family with worse financial well-being (*B* = 0.039, *p* < 0.001), and having unemployed parents (*B* = 0.043, *p* < 0.001) were positively associated with cyberbullying victimization. Of all predictor variables, less computer usage was found to have the most significant effect on cyberbullying victimization.

As shown in Table 3, less computer usage was also significantly and negatively associated with face-to-face bullying victimization (*B* = −0.092, *p* < 0.001), controlling for the demographic variables. Moreover, age (*B* = −0.121, *p* < 0.001) was negatively associated with face-to-face bullying victimization. Adolescents who were Black/African American (*B* = 0.038, *p* < 0.01), White (*B* = 0.045, *p* < 0.01), or American Indian/Alaska Native (*B* = 0.033, *p* < 0.01) were more likely to report face-to-face bullying victimization. Adolescents with better family financial well-being (*B* = 0.068, *p* < 0.001) and those who had two unemployed parents (*B* = 0.033, *p* < 0.01) were also more likely to report face-to-face bullying victimization. The effect sizes for age, a lesser extent of computer usage, and family’s financial well-being were the largest.

## 4. Discussion

The present study investigated how less computer usage might be related to cyberbullying and face-to-face bullying victimization among a large sample of U.S. public school students. A large body of research literature suggests a positive association between material deprivation and victimization [11,24,25], as low SES is a salient risk factor for bullying victimization. However, our findings suggest the opposite in the context of cyberbullying—that is, less computer usage was negatively associated with cyberbullying victimization, which contradicts the income inequality hypothesis, although it supports Hypothesis 1’. As previously stated, the income inequality hypothesis purports that deprivation would be associated with increased inequality, reinforcing problems and conflicts, such as violence [14]. This study’s finding, however, suggests that less computer usage correlates with less indirect bullying, as these individuals are less frequently exposed to cyberspace, where incidents of cyberbullying most often occur.

For the covariates, the study found that cyberbullying victimization was significant for girls, Black/African Americans, families with worse financial well-being, and those with two unemployed parents, which is consistent with other studies’ findings and somewhat supports the income inequality hypothesis [4,9,10,16,17]. Girls may have a significant risk of cyberbullying victimization, as girls tend to communicate using text messaging and email more frequently than boys [34]. Black/African American adolescents might be vulnerable to cyberbullying victimization because they tend to be directly and indirectly bullied [35]. Youths whose families are not well off and those whose parents are unemployed run the risk of cyberbullying victimization, which supports prior study findings that indicate a low SES is a significant risk marker for victimization [16,17]. 

The study also found a negative association between less computer usage and face-to-face bullying victimization, which contradicts prior research [11,24,25] and Hypothesis 2. However, this finding is not surprising in light of Tippett and Wolke’s [15] meta-analytic study, which reported that SES is weakly associated with adolescents’ bullying roles. Material deprivation might not necessarily lead to bullying victimization. Adolescents who are deprived of resources such as a computer also might have limited interactions with their peers in school, potentially shielding them from cyberbullying and face-to-face bullying. 

Regarding the covariates, the study found that face-to-face bullying victimization was significant for adolescents who were Black/African American, White, or American Indian/Alaska Native. Those with better family financial well-being and those who had two unemployed parents were also victims of face-to-face bullying. These findings are somewhat congruous with prior findings [8,9,10,16,17] and suggest that bullying is a severe concern for Black/African American, White, and American Indian/Alaska Native youths who tend to be targeted for their race. Unexpectedly, face-to-face bullying victimization is significant for adolescents with better family financial well-being, contrary to other studies [16,17]. This result might suggest that adolescents whose families present themselves as well off are victimized because of how they are being perceived and resented by their peers. Additional investigation is needed to clarify this mechanism. As expected, however, youths who report that their parents are unemployed have an elevated risk of face-to-face bullying victimization as these youths are likely to come from a family of low SES background. This finding confirms the income inequality hypothesis that low family SES is a significant antecedent of face-to-face bullying. 

These findings highlight the complexities involved in understanding the relationship between SES and bullying victimization. Although the results might shed new light on this relationship, it is important to acknowledge several limitations, including the cross-sectional study design, which inhibited our understanding of causal linkages. A longitudinal study design, which allows for examining how less computer usage during childhood might be associated with adolescents’ cyberbullying and face-to-face bullying victimization, is warranted. The self-report measures of the variables in the study are another limitation, which likely introduced reporting biases. The question about face-to-face bullying victimization and cyberbullying victimization is also problematic in that the question “How often have you been bullied at school in the past couple of months in the ways listed below?” was followed by both face-to-face bullying victimization and cyberbullying victimization. And finally, the HBSC dataset, which is dated, represents another limitation. The study relied on data collected from 2009 to 2010 before the COVID-19 pandemic. However, despite the rapid changes in information and communication technology, the findings in this study provide important insights into the risk and protective factors associated with access to particular resources. According to the National Center for Educational Statistics, between 2019 and 2020, the rate of bullying remained at 22%, and the rate of cyberbullying increased from 8% in 2009 to 16% in 2019 [36]. These rates seem to indicate that bullying and cyberbullying continue to be prevalent problems, and the findings from the current study are still relevant. 

### 4.1. Implications for Future Research

In light of the limitations of the study, future studies should build on the current findings by exploring mediators in the association between computer access and bullying victimization. For example, less computer access might be related to a lower risk of face-to-face bullying and cyberbullying victimization through the mediating role of delinquent peer influence. Future studies might also test whether the routine activities theory can explicate the association between having less computer access and cyberbullying victimization. According to the routine activities theory, cyberbullying victimization is likely to occur through the presence of a motivated offender (e.g., cyberbullies), target suitability (i.e., being perceived as an easy target), and a lack of a capable guardian (e.g., adults) [37]. Adolescents with limited computer usage would not be perceived as easy targets and are less likely to be exposed to a motivated offender (e.g., bullies), making them less at risk of cyberbullying victimization. Future studies might also examine whether the findings are similar when demographic differences, for example, gender and racial/ethnic differences, are considered. Future studies might investigate whether having less access to computers is negatively associated with cyberbullying and face-to-face bullying victimization for racial and ethnic minority adolescents as they could be for White adolescents. 

### 4.2. Implications for Practice

The findings from the study have implications for research and practice. In today’s digital age, adults and youths are largely connected digitally. This study suggests that having less computer usage could potentially act as a protective factor against cyberbullying and face-to-face bullying victimization. For adolescents who have access to computers, parents and teachers are advised to closely monitor and set limitations on adolescents’ use of computers. Although closely monitoring and setting limits on computers may be more of a challenge for older students (e.g., high school students), these adolescents must be made aware that cyberbullying will be taken seriously by adult authorities, such as teachers and school officials. Practitioners must also be prepared to address students’ cyberbullying victimization by working with teachers to address cyberbullying effectively. Practitioners might consider social–emotional learning (SEL) programs, which target students’ social and emotional competencies [38]. SEL programs are designed to improve students’ social skills and lower problematic behaviors [38]. SEL has also demonstrated effectiveness in reducing students’ involvement in bullying [39,40]. Moreover, in 2010, the U.S. Department of Education developed a framework of common components of laws, policies, and regulations focused on school bullying [41]. Since then, U.S. school districts in all 50 states have taken action to have an anti-bullying measure in place [41]. 

Cultural, economic, and societal factors, such as a family’s SES, less computer usage, and sociocultural norms, underlie the associations between less computer usage and race/ethnicity, cyberbullying victimization, and face-to-face bullying victimization. Practitioners need to examine the importance of these contextual differences in explaining the cyberbullying victimization and face-to-face bullying victimization differences found here. Although digital deprivation was negatively associated with cyberbullying and face-to-face bullying victimization in the study, school management teams, practitioners, and policymakers need to collaborate to develop effective prevention programs for socioeconomically disadvantaged adolescents. Also, the present findings suggest the need to consider a gender-specific approach, as girls are at a higher risk of cyberbullying victimization than boys [42]. Since socialization tends to be different for boys and girls, intervention strategies need to be tailored to address issues that are more common among girls, such as gossiping, spreading rumors, and shaming. In terms of implications for policy, in response to the growing concerns, numerous bullying and cyberbullying prevention and intervention programs have been developed and implemented over the years. Despite such efforts, however, the effectiveness of anti-bullying programs in the United States is reported to be modest in comparison to that of programs outside the United States [43].

## 5. Conclusions

The current study findings suggest that more research is needed to untangle the complexities of how deprivation might be related to children’s bullying victimization, both cyber and face-to-face, which could better guide us in developing prevention and intervention efforts targeting disadvantaged children. Even though less computer access was negatively associated with face-to-face bullying and cyberbullying victimization, adolescents who are deprived are at significant risk of bullying victimization [28]. More research on ways in which SES is linked to children’s involvement in bullying is warranted, and this is the first necessary step towards the development of effective anti-bullying prevention and intervention.

## Figures and Tables

**Table 1 behavsci-13-00834-t001:** Descriptive Statistics.

Variables	*n*	%	*M*	*SD*
Face-to-face bullying victimization			5.84	3.15
Cyberbullying victimization			4.58	2.18
Less computer usage			27.38	6.34
Sex				
Boy	6502	51.4%		
Girl	6136	48.6%		
Age			12.95	1.75
Race/Ethnicity				
Hispanic/Latino	3407	28.7%		
Black/African American	2562	20.3%		
White	6581	52.1%		
Asian	681	5.4%		
American Indian or Alaska Native	648	5.1%		
Native Hawaiian or Other Pacific Islander	225	1.8%		
Family’s financial well-being			2.54	0.97
Parents’ occupations				
Does your father have a job?				
No	1007	9.5%		
Yes	9547	90.5%		
Does your mother have a job?				
No	2911	25.1%		
Yes	8672	68.6%		

**Table 2 behavsci-13-00834-t002:** Regression Analyses of the Association Between Less Computer Usage and Cyberbullying Victimization.

Variable	*B*	*SE*	*p*
Less computer usage	−0.086	0.004	0.000 ***
Sex	0.022	0.044	0.041 *
Age	−0.004	0.013	0.689
Ethnicity (Hispanic/Latino)	−0.050	0.065	0.000 ***
Race			
Black/African American	0.039	0.075	0.005 **
White	0.017	0.064	0.263
Asian	−0.028	0.101	0.018 *
American Indian or Alaska Native	0.002	0.102	0.858
Native Hawaiian or Other Pacific Islander	0.007	0.167	0.550
Family’s financial well-being	0.039	0.024	0.000 ***
Parents’ occupation	0.043	0.042	0.000 ***
R square	0.016		
Adjusted R square	0.015		

*** *p* < 0.001, ** *p* < 0.01, * *p* < 0.05.

**Table 3 behavsci-13-00834-t003:** Regression Analyses of the Association Between Less Computer Usage and Face-to-Face Bullying Victimization.

Variable	*B*	*SE*	*p*
Less computer usage	−0.092	0.005	0.000 ***
Sex	0.014	0.066	0.190 ^NS^
Age	−0.121	0.020	0.000 ***
Ethnicity (Hispanic/Latino)	−0.023	0.097	0.105 ^NS^
Race			
Black/African American	0.038	0.112	0.005 **
White	0.045	0.095	0.003 **
Asian	−0.014	0.151	0.242 ^NS^
American Indian or Alaska Native	0.033	0.150	0.002 **
Native Hawaiian or Other Pacific Islander	0.016	0.248	0.146 ^NS^
Family’s financial well-being	0.068	0.035	0.000 ***
Parents’ occupation	0.033	0.062	0.002 **
R square	0.031		
Adjusted R square	0.029		

*** *p* < 0.001, ** *p* < 0.01, NS = not significant.

## Data Availability

Not applicable.

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
