# Peer review of "Less Computer Access: Is It a Risk or a Protective Factor for Cyberbullying and Face-to-Face Bullying Victimization among Adolescents in the United States?"

_behavsci, 2023, doi:10.3390/bs13100834_

Round 1
Reviewer 1 Report
Overall, I thought the article was well done but there are some critical parts that need to be included:
1) The gap in the literature should be explicitly stated
2) The author(s) need address their hypotheses in the discussion. Were their hypotheses supported? I saw this briefly discussed-may just need to be reworded.
3) Also, the authors may consider adding another hypothesis that is more specific to the gap in the literature. This allows the reader to clearly understand that this research is important and fills a gap.
4) Make sure all references are current as research on bullying and cyberbullying is constantly changing and expanding
5) Make sure the discussion can be directly related to the introduction/literature review.
No major issues.
Author Response
Review #1:
Overall, I thought the article was well done but there are some critical parts that need to be included:
- The gap in the literature should be explicitly stated
We stated the following: “To date, a bulk of empirical studies have focused on whether poverty and deprivation are related to children’s experiences of face-to-face bullying. Additionally, research has not considered whether deprivation, such as a lack of computer access, for example, might be protective against bullying and cyberbullying as deprivation can shield youth from having contact with bullies.”
- The author(s) need address their hypotheses in the discussion. Were their hypotheses supported? I saw this briefly discussed-may just need to be reworded.
We wrote “although it supported our first hypothesis” in the first paragraph of the Discussion section. In the third paragraph, we wrote “contradicts prior research [11, 24, 25] and our second proposed hypothesis”.
- Also, the authors may consider adding another hypothesis that is more specific to the gap in the literature. This allows the reader to clearly understand that this research is important and fills a gap.
Thank you very much for this insight. However, we felt that the first hypothesis was specific to the gaps in the literature in which we stated, “Additionally, research has not considered whether deprivation, such as a lack of computer access, for example, might be protective against bullying and cyberbullying as deprivation can shield youth from having contact with bullies.”
4) Make sure all references are current as research on bullying and cyberbullying is constantly changing and expanding
The references are current and we were very careful about citing the most relevant literature.
5) Make sure the discussion can be directly related to the introduction/literature review.
Reviewer 2 Report
Dear authors, I had the pleasure of reading your paper but I have some important considerations to make.
Please pay attention in the language used. Sometimes authors refers to cybervictimization using " bullying victimization" this means traditional bullying victimization. Please check carefully and replace the terms.
Please check carefully all sentences, some words are missed. E.g. "The present utilized data from..." replace with "The present study/research etc..."
It is necessary to increase subparagraph 1.2 to better explain the need to investigate the relationship between access to electronic resources and low involvement in cybervictimization, reporting previous studies.
I think there is an error in the wording of the questions in the questionnaire, but it is a problem that can no longer be solved. It doesn't seem correct to ask the same question "How often have you been bullied at school in the past couple of months in the ways listed below?" with reference to two different phenomena (bullying and cyberbullying) although there are completely different answers. This should be mentioned among the limitations of the study.
Your results report that less computer use is negatively associated with less cybervictimization which is not in line with what you state in the discussions, you affirm that your study's results suggests that less computer usage correlates with less indirect bullying . Please check carefully.ùùIn general, the data are certainly interesting despite the fact that they refer to non-current research, but they must be discussed more carefully.
Author Response
Review #2:
Dear authors, I had the pleasure of reading your paper but I have some important considerations to make.
Thank you very much for your very insightful comments and suggestions.
Please pay attention to the language used. Sometimes authors refers to cybervictimization using " bullying victimization" this means traditional bullying victimization. Please check carefully and replace the terms.
Thank you very much. We made sure that the correct terminology was being used.
Please check carefully all sentences, some words are missed. E.g. "The present utilized data from..." replace with "The present study/research etc..."
Thank you very much. We checked to make sure there weren’t any missed words.
It is necessary to increase subparagraph 1.2 to better explain the need to investigate the relationship between access to electronic resources and low involvement in cybervictimization, reporting previous studies.
We included the following paragraph “To date, a bulk of empirical studies have focused on whether poverty and deprivation are related to children’s experiences of face-to-face bullying. Additionally, research has not considered whether deprivation, such as a lack of computer access, for example, might be protective against bullying and cyberbullying as deprivation can shield youth from having contact with bullies”, which is located prior to “The Present Study”.
I think there is an error in the wording of the questions in the questionnaire, but it is a problem that can no longer be solved. It doesn't seem correct to ask the same question "How often have you been bullied at school in the past couple of months in the ways listed below?" with reference to two different phenomena (bullying and cyberbullying) although there are completely different answers. This should be mentioned among the limitations of the study.
In the limitations section, we mentioned the following: “The question about face-to-face bullying victimization and cyberbullying victimization is also problematic in that the question, “How often have you been bullied at school in the past couple of months in the ways listed below?” was followed by both face-to-face bullying victimization and cyberbullying victimization.”
Your results report that less computer use is negatively associated with less cybervictimization which is not in line with what you state in the discussions, you affirm that your study's results suggest that less computer usage correlates with less indirect bullying. Please check carefully. In general, the data are certainly interesting despite the fact that they refer to non-current research, but they must be discussed more carefully.
We apologize for the confusion. We made the following correction in the discussion section: “However, our findings suggest the opposite in the context of cyberbullying—that is, less computer usage was negatively associated with cyberbullying victimization, which contradicted the income inequality hypothesis although it supported our first hypothesis.”
Round 2
Reviewer 1 Report
Overall, much better. My only suggestion is to proofread one more time to check for any grammatical issues.
Reviewer 2 Report
dear Authors, I appreciated your efforts to improve your paper